# Genome-Wide Copy Number Variant and High-Throughput Transcriptomics Analyses of Placental Tissues Underscore Persisting Child Susceptibility in At-Risk Pregnancies Cleared in Standard Genetic Testing

**DOI:** 10.3390/ijms231911448

**Published:** 2022-09-28

**Authors:** Darina Czamara, Cristiana Cruceanu, Marius Lahti-Pulkkinen, Linda Dieckmann, Maik Ködel, Susann Sauer, Monika Rex-Haffner, Sara Sammallahti, Eero Kajantie, Hannele Laivuori, Jari Lahti, Katri Räikkönen, Elisabeth B. Binder

**Affiliations:** 1Department of Translational Research in Psychiatry, Max-Planck-Institute of Psychiatry, 80804 Munich, Germany; 2Department of Physiology and Pharmacology, Karolinska Institutet, 171 77 Stockholm, Sweden; 3Department of Psychology and Logopedics, Faculty of Medicine, University of Helsinki, 00014 Helsinki, Finland; 4Finnish Institute for Health and Welfare, 00271 Helsinki, Finland; 5Centre for Cardiovascular Science, Queen’s Medical Research Institute, University of Edinburgh, Edinburgh EH16 4TJ, UK; 6International Max Planck Research School for Translational Psychiatry, 80804 Munich, Germany; 7Department of Obstetrics and Gynaecology, Helsinki University Hospital and University of Helsinki, 00014 Helsinki, Finland; 8Children’s Hospital, Helsinki University Hospital and University of Helsinki, 00014 Helsinki, Finland; 9Faculty of Medicine, PEDEGO Research Unit, MRC Oulu, Oulu University Hospital and University of Oulu, 90014 Oulu, Finland; 10Department of Clinical and Molecular Medicine, Norwegian University of Science and Technology, 7491 Trondheim, Norway; 11Institute for Molecular Medicine Finland, HiLIFE, University of Helsinki, 00014 Helsinki, Finland; 12Medical and Clinical Genetics, University of Helsinki and Helsinki University Hospital, 00014 Helsinki, Finland; 13Department of Obstetrics and Gynecology, Tampere University Hospital and Faculty of Medicine and Health Technology, Center for Child, Adolescent and Maternal Health Research, Tampere University, 33520 Tampere, Finland; 14Department of Psychiatry and Behavioral Sciences, School of Medicine, Emory University, Atlanta, GA 30322, USA

**Keywords:** chorionic villus sampling, congenital malformations, placenta, prenatal testing transcriptome sequencing

## Abstract

Several studies have shown that children from pregnancies with estimated first-trimester risk based on fetal nuchal translucency thickness and abnormal maternal serum pregnancy protein and hormone levels maintain a higher likelihood of adverse outcomes, even if initial testing for known genetic conditions is negative. We used the Finnish InTraUterine cohort (ITU), which is a comprehensively characterized perinatal cohort consisting of 943 mothers and their babies followed throughout pregnancy and 18 months postnatally, including mothers shortlisted for prenatal genetic testing but cleared for major aneuploidies (cases: n = 544, 57.7%) and control pregnancies (n = 399, 42.3%). Using genome-wide genotyping and RNA sequencing of first-trimester and term placental tissue, combined with medical information from registry data and maternal self-report data, we investigated potential negative medical outcomes and genetic susceptibility to disease and their correlates in placenta gene expression. Case mothers did not present with higher levels of depression, perceived stress, or anxiety during pregnancy. Case children were significantly diagnosed more often with congenital malformations of the circulatory system (4.12 (95% CI [1.22–13.93]) higher hazard) and presented with significantly more copy number duplications as compared to controls (burden analysis, based on all copy number variants (CNVs) with at most 10% frequency, 823 called duplications in 297 cases versus 626 called duplications in 277 controls, *p* = 0.01). Fifteen genes showed differential gene expression (FDR < 0.1) in association with congenital malformations in first-trimester but not term placenta. These were significantly enriched for genes associated with placental dysfunction. In spite of normal routine follow-up prenatal testing results in early pregnancy, case children presented with an increased likelihood of negative outcomes, which should prompt vigilance in follow-up during pregnancy and after birth.

## 1. Introduction

Prenatal screening in early pregnancy is instrumental in monitoring maternal health and proper child development. Usually, women at risk for carrying babies with major aneuploidies are identified using an assessment of risk by a combined first-trimester screening battery that includes: fetal nuchal translucency (NT) thickness, maternal levels of serum pregnancy-associated plasma protein A (PAPP-A) and free β-human chorionic gonadotropin (β-hCG) [1] in addition to maternal age, maternal weight and gestational week. Although this screening test has proven to successfully identify risk for the major chromosomal aneuploidies, several studies have shown that fetuses with elevated risk at screening but cleared for the major aneuploidies in prenatal genetic testing remain at higher risk for negative outcomes diagnosed before or after birth [2,3,4]. Bardi et al. reported that 34% of congenital malformation cases would be missed if prenatal testing only relied on the standard screening for major aneuploidies [2]. Zhang et al. [4] showed that significantly more clinically relevant pathogenic copy number variants (CNVs) are present in fetuses with NT between 2.5 mm and 3.5 mm, which is below the commonly used threshold of 3.55 mm. Furthermore, low PAPP-A serum levels in the first trimester of pregnancy have been associated with short stature in offspring and de-novo development of maternal diabetes mellitus later in life [3]. This suggests that using these values on a continuum rather than as a threshold in the context of other indicators of risk might provide added value to the current clinical practices and detect individuals with underlying disease susceptibility otherwise not identified in the prenatal period.

Chorionic villus sampling (CVS), amniocentesis, or noninvasive prenatal testing (NIPT) of fetal DNA in combination with 11-to-13-week ultrasound examination, can detect major negative developmental outcomes with 70–100% detection accuracy [5,6]. Invasive approaches have been available since the 1970s [7], while noninvasive approaches have only become routine practice in developed countries in the last 10 years [6]. The latter have rapidly developed into the method of choice, given the much lower risk for mother and fetus compared to invasive techniques, and are being administered in more than half of pregnancies in developed countries [5]. However, NIPT is a screening test, and invasive techniques are still considered state-of-the-art in validating suspicious noninvasive results [6]. 

We set out to understand if children from pregnancies flagged for potential chromosomal aberrations but ultimately cleared for the major aneuploidies following CVS or amniocentesis differed from children of inconspicuous pregnancies and how detection of prevailing pathologies could be improved. To explore this question, we used a thoroughly characterized perinatal cohort, the Finnish InTraUterine cohort (ITU), consisting of 943 mothers and babies followed throughout pregnancy and postnatally until 18 months. We used genome-wide high-throughput analyses at the level of RNA and DNA methylation in cord blood, term placentas, and, to the best of our knowledge, in the largest sample to date of CVS biopsies in the subset of pregnancies in which chromosomal testing was performed. Our findings point to an enduring potential for concern for case children, with increased risk for congenital malformations. 

## 2. Results

### 2.1. Case Mothers Do Not Differ from Controls in Psychological State across Pregnancy following Trisomy Screening Clearance

As shown by Kvist et al. [8], mothers shortlisted for prenatal screening (cases, n = 544) significantly differed from control mothers (n = 399): they were older, presented with higher body mass index (BMI), had had more pregnancies and smoked more often. By design, cases also significantly differed from controls with regards to NT, PAPP-A, and β-hCG levels as well as risk for Down syndrome or trisomy 18. The main characteristics of the cohort and of the screening variables are presented in Table 1. 

Furthermore, we assessed whether maternal psychological state differed in relation to case status, i.e., if mothers who had undergone prenatal testing possibly presented with higher levels of depression, anxiety or perceived stress due to the testing or its possible implications for the health of their child. Questionnaires on well-being throughout pregnancy were completed by 613 women, at three time points roughly representative of the three trimesters. We found no significant differences between cases and controls in any of the assessments during pregnancy (see Table 2) indicating that mothers did not differ in their psychological state across pregnancy.

### 2.2. Children from Pregnancies Recommended for Follow-Up Prenatal Genetic Testing Carry Higher Likelihood of Negative Developmental and Disease-Related Outcomes

#### 2.2.1. Congenital Malformations

We examined whether cases and controls differed with regards to negative medical outcomes and genetic susceptibility, starting with congenital malformations, as the first indication of underlying disease risk. Of 943 children, 98 (68 cases, 12.5% and 30 controls, 7.5%) presented with any congenital malformations diagnosed up until 2017. We ran Cox regression models to account for different diagnostic follow-up duration in the Finnish nationwide Care Register for Healthcare (HILMO) for children born in different years. Cox regression models showed no difference between groups in all malformation types (*p* = 0.14, hazard ratio = 1.39, 95% CI [0.47–1.11]). However, in circulatory system congenital malformations comprising 25 diagnoses (22 cases, 4.0%, and 3 controls, 0.8%), we found a 4.12 (95% CI [1.22–13.93]) higher hazard in the cases (*p* = 0.02, see Figure 1). The majority of affected children (n = 17, 65.4%) presented with ventricular septal defects. Beyond malformations of the circulatory system, there were some isolated cases of malformations of the respiratory, musculoskeletal, genital, digestive, or nervous systems that were too infrequent to permit statistical analysis.

In a next step, we investigated if congenital malformations of the circulatory system were specifically associated with any of the screening variables. While there were no significant associations with higher NT, higher β-hCG levels or higher risk for trisomy 18, children with congenital malformations of the circulatory system presented with significantly higher estimated ratio for Down syndrome and lower, albeit not significantly different, PAPP-A levels. These findings remained stable, even after correcting for case-control status, which was itself defined using the screening variables (risk for Down syndrome: *p* = 0.021, PAPP-A level: *p* = 0.058).

#### 2.2.2. Copy Number Variants

Given the importance of CNVs in disease susceptibility, including in association with congenital malformations [10], and the association of sub-threshold screening results with a higher number of CNVs [4], we next evaluated CNVs in our sample. We first called CNVs across all participants. The large majority of CNVs, over 92%, was classified as not pathogenic (see Methods). We found no significant differences between individuals with or without congenital malformations (any or circulatory system specific), neither on single CNV level nor based on burden analysis. This is possibly due to a reduced detection power given the low number of congenital malformations. Since significantly more clinically relevant pathogenic CNVs have been associated with risk-level prenatal screening variables such as maternal age and NT abnormalities [4,11], we also tested this association at the level of the entire cohort. In our cohort, cases presented with overall more duplications, but not more deletions, as compared to controls (burden analysis, based on all CNVs with at most 10% frequency, 823 called duplications in 297 cases versus 626 called duplications in 277 controls, *p* = 0.01, based on 10,000 permutations, see Figure 2). This range of CNV counts is in line with previous studies using similar technology and sample sizes [12,13].

### 2.3. The Placental Transcriptome in Early Pregnancy Carries Signatures of Case-Control-Associated Negative Developmental Outcomes

Given the observed differences in congenital malformations within the circulatory system and in CNVs, we next assessed if these were also associated with RNA expression in early and late placental tissue. Gene expression derived from CVS tissue was available for 266 case mothers who had undergone invasive prenatal testing via placental biopsies in the first trimester. We found no differentially expressed genes to associate with CNVs (number of duplications). However, 15 genes were significantly differentially upregulated in individuals with circulatory system congenital malformations (10 individuals with and 256 without diagnosed congenital malformations; see Figure 3 and Table 3).

Correlation of gene expression for these 15 genes between CVS and term placenta (based on 93 individuals who had both tissues available) revealed significant, albeit weak to moderate, positive correlations for 7 of these transcripts (r = 0.23 to r = 0.30, see Table 3). Of these 15 genes, 11 had previously been shown to be dysregulated in pre-eclampsia [14,15,16]. This is significantly more than expected by chance (*p* = 9.91 × 10^−11^, see Methods). Importantly, it should be noted that only one case presenting with congenital malformations of the circulatory system also had a pre-eclampsia diagnosis. Removal of this case did not change the effect direction of the 15 genes.

In term placenta, we found no significantly differentialy expressed genes with circulatory system congenital malformations. However, it should be noted that only six individuals with this phenotype and term placenta gene expression were available hence this null result may be due to power issues.

Number of CNV duplications was associated with differential placental gene expression of *RFLNB* (based on 433 samples with information on CNVs and placental gene expression, log2(FC) = 0.07, adjusted *p*-value = 3.72 × 10^−2^, correlation with CVS gene expression r = 0.16, *p*-value = 0.13). This result stayed significant after accounting for case-control status with regard to prenatal screening.

## 3. Discussion

Advances in medical prenatal screening and care during early pregnancy have led to life-saving effects, with infant mortality decreasing steadily over the past two decades, by as much as three-fold in high-income countries and by half globally [17]. However, there are still improvements to be made regarding access to state-of-the-art prenatal screening, as well as improved interpretation of available diagnostic tools. While sampling of CVS tissue or amniotic fluid to test the fetal DNA for major chromosomal rearrangements has been the gold standard for many decades, there is a significant risk for miscarriage linked with these invasive techniques [6]. In addition, they could lead to false positive or false negative outcomes in addition to sampling inaccuracy. Noninvasive prenatal screening techniques based on whole-genome DNA sequencing have become increasingly common and accurate and represent a suitable companion or replacement technique since they can detect cell-free fetal DNA in the mother’s blood that is released from fetal placenta apoptotic trophoblasts [6]. However, a major limitation of this approach is sensitivity, given that circulating fetal DNA constitutes only a small fraction of the mother’s blood [6]. Nonetheless, genetic screening, in addition to combined first-trimester testing for fetal NT and maternal PAPP-A and βhCG, has been very powerful in detecting individuals susceptible to congenital disease. With this work, we show that beyond indicating risk for major aneuploidies, common prenatal screening measurements can indicate the presence of additional risk for disease.

An interesting resource for improving prenatal or early-life diagnostics is the incorporation of perinatal tissues such as the placenta and cord blood into the interpretation of congenital and genetic risk of disease. In this manuscript, we investigated a case-control cohort of 943 mother-baby dyads whereby the case mothers (n = 544) were initially screened as having an increased risk for chromosomal abnormalities, but ultimately the children were deemed normal in regard to major aneuploidies following invasive genetic testing or NIPT. We compared these to a group of controls without any risk indicator at screening (n = 399). We found significant differences at the level of perinatal child outcomes regarding congenital malformations of the circulatory system, as well as CNV load. In addition, we found an association between child outcomes and first-trimester but not term placenta gene expression signatures. Interestingly, there was no difference in self-reported maternal stress or anxiety during pregnancy, in spite of the case mothers being exposed to the stressful experience of invasive prenatal testing and anticipation of results. However, it bares noting that there was a disproportionately lower adherence in the case mothers regarding the psychological self-assessment during pregnancy, which might influence the outcome of these analyses. Future studies should assess the impact of a similar exposure in additional cohorts.

We found an increased likelihood of a diagnosis of circulatory system congenital malformation for case children, which supports previous research linking abnormal NT results at screening with increased rates of congenital malformations and heart disease [2]. However, this study is the first to show that the congenital malformations phenotype is associated with gene expression in the first-trimester placenta from CVS sampling. A total of 15 genes showed differential expression association with circulatory system congenital malformations in first-trimester placenta obtained from CVS. Correlations with gene expression in placentas sampled at birth were weak to moderate, so we cannot directly translate our findings to late pregnancy. Of these 15 genes, 11 had previously been shown to be dysregulated in pre-eclampsia [14,15,16], significantly more than expected by chance.

A number of the genes differentially expressed in association with congenital malformations in the CVS tissue have previously been shown to be important mediators of placenta biology and associated with pregnancy diseases. The top-ranking gene by fold change, Leptin (*LEP*), is a circulating hormone involved in metabolism and energy homeostasis [18] and secreted primarily from adipose tissue but also from placenta cells such as cytotrophoblasts, syncytiotrophoblasts, and villous vascular endothelial cells. This hormone is upregulated in the placentas of mothers with pregnancy complications such as pre-eclampsia [19], which is the same direction of effect we found for early placentas from children with underlying congenital malformations. Secondly, corticotropin-releasing hormone (*CRH*) is a major mediator of hypothalamic–pituitary–adrenal axis (HPA axis) responses to stress and a key regulator of brain development. In addition to the paraventricular nucleus (PVN) of the hypothalamus, CRH is also synthesized by the placenta (pCRH) as early as post-conceptional week 7 and is found in both the maternal and the fetal compartments [20]. Down-regulation of this gene at the RNA level has been linked to placenta inflammation, while up-regulation, as in the case of our analyses, has been linked to motor dysfunctions [21] in children and pre-eclampsia in pregnancy [22]. Thirdly, elevated levels of the follistatin-like 3 protein (*FSTL3*) gene, which encodes a protein secreted by syncytiotrophoblast cells, have been linked to pre-eclampsia by several studies. In fact, placentally-derived FSTL3 detected in the maternal serum has been proposed as a third-trimester diagnostic biomarker of pre-eclampsia [23]. The next differentially expressed genes by fold change were pregnancy-associated plasma protein A2 (*PAPPA2*) [24], inhibin beta A subunit (*INHBA*) [25], and HtrA serine peptidase 1 (*HTRA1*) [26]. All these genes, in addition to several others identified through our analyses, were previously shown to be elevated at the mRNA and protein level in pre-eclampsia with severe features, early-onset pre-eclampsia, or fetal growth restriction. These findings may suggest that first-trimester placental tissue of fetuses with congenital anomalies presents with signatures of placental dysfunction with a possible additional impact on fetal health. Interestingly this signature disappeared with the maturation of the placenta, and there was no increase in rates of pre-eclampsia diagnoses.

Given the indication of existing susceptibility to disease in the early fetal placental tissue, we used child DNA extracted from cord blood to further investigate complex genetics that would not have been detected by prenatal genetic screening. We focused on CNVs, given their association with both congenital malformations [10] and general pregnancy risk factors such as advanced maternal age [11]. In addition to the previously suggested associations of increased numbers of pathogenic CNVs with positive prenatal screening [4], we demonstrated a novel connection whereby the cases presented with more duplications overall. Together these findings suggest an overall increase in potentially harmful genetic loading in the case children. Importantly, this detail of genetic variation could not be detected with currently available prenatal genetic screening methods due to limited fetal DNA amounts available at early stages.

The analyses described in this manuscript identified significant connections between first-trimester placenta gene expression (CVS) and child outcomes, specifically regarding congenital malformations. Conversely, associations to the number of CNV duplications were only identified in the term placenta RNA, related to the gene *RFLNB*, which is enriched in Hofbauer cells and previously associated with early signs of pre-eclampsia in the placental transcriptome [27]. This apparent discrepancy between first-trimester placentas from CVS and term placentas could be due to different reasons. Primarily, samples with CVS and term placenta tissue only partly overlap, and hence we cannot make robust conclusions on the individual developmental trajectories. In fact, only one individual with a circulatory system congenital malformation diagnosis also had both tissues available. However, having access to early pregnancy tissues, as in the ITU cohort, is extremely valuable in establishing the connections between placenta biology and underlying child disease risk. As mentioned above, our data may point to additional early placental dysfunction in children with congenital malformations of the circulatory system. Functional studies of early placental physiology could shed more light on the possible clinical relevance of this finding.

Some of the limitations of this study highlight the need for even larger, well-characterized longitudinal cohorts, where these findings can be independently replicated and extended with measures of placental function. Importantly, extended follow-up of the children will undoubtedly clarify and complete the picture of potential negative outcomes in this at-risk population. The ITU cohort, with the ongoing characterization of participants in addition to Finnish health registry data, will constitute a rich resource moving forward.

The findings identified in this study join other reports suggesting that children with elevated risk but not meeting formal diagnostic thresholds during prenatal screening maintain a higher likelihood of negative outcomes. Increased awareness is key at the level of medical professionals as well as the general public with regard to potential early indicators of underlying disease susceptibility and the prevention or intervention measures that can be taken to improve the outcome and quality of life of affected individuals. In our cohort, cases were significantly older as compared to control women, which is in line with what also Hayeems et al. reported [28]. Women of higher age are more likely to undergo prenatal testing, and higher maternal age has been associated with a higher risk for fetal aneuploidies [29]. Given that maternal age at first birth is shifting to older ages [30], this will likely be an issue of increasing importance. Our findings point to a need for redefining the risk classifications in relation to prenatal screening and more vigilance during prenatal and postnatal follow-up of at-risk children.

## 4. Materials and Methods

### 4.1. Sampling and Phenotypes

#### 4.1.1. Study Samples

The InTraUterine sampling in early pregnancy (ITU) cohort consists of 943 Finnish women and their children born between 2012 and 2017. The cohort, described by Kvist et al. [8], is a prospective, longitudinal pregnancy cohort study comprising a total of 943 women. Pregnant women were recruited at maternity clinics and through the Helsinki and Uusimaa Hospital District Fetomaternal Medical Center in Finland. Eligibility criteria included singleton pregnancy, no prenatal diagnosis of chromosomal abnormality, maternal age ≥ 18 years, and sufficient Finnish language ability to ensure informed consent. The ITU study comprises two study arms. Women in the chromosomal testing arm (cases, n = 544) had been referred to the Helsinki and Uusimaa Hospital District Fetomaternal Medical Center (FMC) because they had an increased risk of fetal chromosomal abnormalities based on routine serum and ultrasound screening, age, and patient characteristics. The screening is described in detail in Kvist et al. [8]. In brief, the screening program was a combination of serum screening and ultrasound examinations, including a nuchal translucency scan. Women who had a positive screening result (i.e., an estimated risk of fetal chromosomal abnormality >1:250, based on serum and ultrasound screening, age, maternal height and weight, and prior history) were then offered fetal chromosomal testing (CVS or amniocentesis followed by trisomy PCR, or noninvasive prenatal testing) at FMC. If the chromosomal test indicated no fetal chromosomal abnormalities, those who had expressed interest in participating were contacted for final recruitment. Those whose chromosomal test results suggested a fetal chromosomal abnormality were not recruited. Women in the no-chromosomal testing arm (controls, n = 399) were informed about ITU when attending the same routine serum and ultrasound screening at maternity clinics as the women in the chromosomal testing arm. Women who expressed interest in participating were contacted for final recruitment into this study arm if they had not been referred to FMC for fetal chromosomal testing. After careful inspection, one woman was excluded from the analyses post hoc as she did not meet the clear-cut case-control definition. One woman who had originally been accidentally coded as control was recoded as case. This resulted in 399 controls and 544 case women in the final analysis. Demographic information on the ITU cohort, as well as on the screening variables, is presented in Table 1.

#### 4.1.2. Phenotypes

##### Maternal Characteristics

Maternal characteristics are described in detail by Kvist et al. [8] and were extracted from the Finnish Medical Birth Register (FMBR) as well as through self-report questionnaires on their depressive, anxiety, and perceived stress symptoms up to three times during pregnancy. With regards to psychometric assessments, the mothers completed questionnaires on their depressive, anxiety, and perceived stress symptoms up to three times during pregnancy, on average (median) at 19.3 [Standard Deviation (SD) = 3.7], 26.1 (SD = 3.1), and 38.1 (SD = 8.5) weeks of gestation, respectively. Depressive symptoms were assessed with the Center for Epidemiologic Studies Depression Scale [31], perceived stress symptoms with a 5-item version of the Cohen’s Perceived Stress Scale [32], and anxiety symptoms with the Spielberger State Anxiety Inventory (STAI) [33]. All three are validated questionnaires [33], and the CESD and the STAI have also been validated among pregnant women [34]. In our sample, the Cronbach’s alphas indicating the internal consistencies of the scales ranged from 0.87 to 0.89 for CESD, from 0.94 to 0.95 for STAI, and from 0.69 to 0.71 for Cohen’s Perceived Stress Scale.

##### Child Characteristics

Data on the child’s sex (girl/boy), gestation length (weeks), and birth weight (grams) were extracted from the FMBR. We extracted data on diagnoses of congenital malformations and, more specifically, on malformations of the circulatory system from the Finnish nationwide Care Register for Healthcare (HILMO). The HILMO healthcare register includes primary and subsidiary diagnoses of all hospitalizations in Finland since 1969 and of all visits in specialized outpatient care since 1998. Diagnoses have been entered into the HILMO according to the International Statistical Classification of Diseases and Related Health Problems, Eighth Revision (ICD-8) until 1986, according to ICD-9 from 1987 to 1995, and according to ICD-10 since 1996. The HILMO is a validated tool for research [35]. We had lifetime data from HILMO available until 31 December 2017. Diagnosis of any congenital malformation, deformation, or chromosomal abnormalities was identified with ICD-10 diagnostic codes Q00-Q99. We also identified congenital malformations, specifically of the circulatory system, with diagnostic codes Q20-Q28. These diagnoses were identified from childbirth until 31 December 2017, when the children were between 2 days and 5.7 years old.

### 4.2. Biosampling, DNA/RNA Extractions

Placenta samples from the fetal side of the placenta, relatively close to the umbilical cord, were collected after birth by midwives who took nine-site biopsies within 120 min of delivery. The biopsies were stored in RNA storage solution (RNAprotect, Qiagen) until frozen by research staff at −80 °C (within 24 h of delivery) for long-term storage. Chorionic villus (CVS) biopsies were collected by experienced obstetricians in early pregnancy (weeks 8–12). After chromosomal analysis, surplus tissue was immediately stored at −80 °C. DNA and RNA were extracted from CVS and delivery placenta, and DNA was extracted from cord blood leukocytes using a bead-based method optimized by tissue type (Chemagic 360 Perkin Elmer). Total CVS biospecimens were homogenized and split 40–60% for RNA-DNA extraction. Delivery placenta samples preserved in RNAprotect reagent were thawed, and equal-sized aliquots were dissected, homogenized, and split 40–60% for RNA-DNA extraction. Quantification and quality assessments were performed using a TapeStation Automated Electrophoresis system (Agilent) and an Epoch Microplate Spectrophotometer (BioTek, Agilent). All extractions were performed at the BioPrep core unit, Max Planck Institute for Psychiatry.

### 4.3. Genotyping, Imputation, and MDS Components

Genotyping was performed on Illumina GSA-24v2-0_A1 arrays, according to the manufacturer’s guidelines (Illumina Inc., San Diego, CA, USA). Quality control is described in detail in Dieckmann et al. [36]. After quality control, genotypes from 592 individuals and 338,132 SNPs were subjected to imputation. Imputation was performed using *shapeit2* [37] and *impute2* [38] based on the 1000 Genomes Phase III reference sample. After imputation, SNPs with an info score below 0.6, a minor allele frequency below 0.01, or deviating from Hardy–Weinberg equilibrium (*p*-value < 1 × 10^−5^) were excluded from further analysis resulting in 9,826,011 SNPs. Multi-dimensional scaling (MDS) was performed in *PLINK* on the genotyped dataset after linkage disequilibrium pruning.

### 4.4. CNV Calling

Quality control on individuals’ SNP genotyping was performed as described in Dieckmann et al. [36]. Briefly, individuals presenting with callrates < 98% or being outliers in the multi-dimensional scaling analysis were removed from the analysis. For calling of CNVs, raw .idat files of these 592 IDs were converted into vcf files and tabular input files using *gtc2vcf* (https://github.com/freeseek/gtc2vcf, accessed on 9 July 2021), *VCF-simplify* (https://github.com/everestial/VCF-simplify, accessed on 9 July 2021)*,* and R [39]. SNPs were excluded if any of these conditions were met: callrate < 98%, cluster separation < 0.3, AB R Mean <= 0.2, AB R Mean <= 0.2, BB R Mean <= 0.2, AB T Mean <= 0.1 or AB T Mean > 0.9, Het Excess < −0.9 or Het Excess > 0.9, minor allele frequency > 0 and AB Freq = 0, AA Freq = 1 and AA T Mean > 0.3, AA Freq = 1 and AA T Dev > 0.06, BB Freq = 1 and BB T Mean < 0.7, BB Freq = 1 and BB T Dev = 0.06. These values were chosen with regard to Illumina’s recommendation (https://www.illumina.com/Documents/products/technotes/technote_infinium_genotyping_data_analysis.pdf, accessed on 9 July 2021). Afterward, CNVs were called using *PennCNV* [40], correcting for GC content as described in Diskin et al. [41]. Adjacent calls were merged and individuals with poor quality parameters were removed, according to the default settings (fraction < 0.2, LRR SD > 0.3, BAF drift > 0.01, WF > 0.05). Furthermore, CNV calls spanning less than 10 SNPs, individuals with more than 100 CNVs, and spurious CNVs, which are likely in centromeric and telomeric regions, were removed according to the recommendations of Lin et al. [42] and Li et al. [43] This resulted in a final sample size of 574 IDs (277 controls and 297 cases) and 9334 detected CNV calls including 3445 CNVs (2290 deletions and 1155 duplications). We used *ClassifyCNV* (https://github.com/Genotek/ClassifyCNV, accessed on 13 July 2021)*,* to classify detected CNVs according to guidelines [44] of the American College of Medical Genetics and Genomics into the categories: benign, likely benign, uncertain significance, likely pathogenic, or pathogenic. Of the called CNVs, 3382 could be defined with *ClassifyCNV*, 239 CNVs were identified as likely pathogenic or pathogenic, 1177 as benign and 1966 with uncertain significance. A detailed list of classified CNVs is given in Appendix A.

### 4.5. RNA Sequencing

The QuantSeq 3′ mRNA-Seq Library Prep Kit (Lexogen) was used to generate messenger RNA (mRNA) sequencing libraries from both term placenta and CVS RNA samples. All libraries were multiplexed and sequenced on an Illumina HighSeq4000 system at a depth of 10 million reads per mRNA library. Adapter sequences were trimmed using cutadapt (https://cutadapt.readthedocs.io/en/stable/), and sequenced reads were aligned to the human genome reference using the STAR aligner [45]. We performed featureCounts [46] and filtered the datasets to genes presenting with a raw count of at least 10 in at least 90% of the individuals, resulting in a final dataset including 8245 transcripts and 493 individuals for the term placentas. For the CVS dataset, the same filtering led to 9089 transcripts quantified in 266 individuals.

### 4.6. Statistical Analysis

All statistical analyses were performed in R version 4.0.4 and SPSS 28.0. *p*-value thresholds are given separately for each sub-analysis.

#### 4.6.1. Differences in Congenital Malformations Using Cox Regression

We checked if cases and controls nominal significantly differed with regards to the hazard of congenital malformations using Cox regression models as implemented in the R-package *survival* 3.2.13. The follow-up time of cases was significantly longer as compared to controls (*p* = 5.52 × 10^−86^, Wilcoxon-test); hence, we limited the follow-up time of cases to the age of the eldest child in the control group and individuals who had been diagnosed with congenital malformations after that time, were set to not diagnosed.

First, we tested which covariates that were different between cases and controls were also significantly nominally associated with congenital malformations using Cox regression models. These covariates (none for any congenital malformation, maternal BMI in early pregnancy for congenital malformations within the circulatory system: *p* = 0.004, hazard ratio = 1.10 [1.03–1.18]) were then included in the final Cox regression model on case-control differences. Survival curves were plotted using the R-package *survminer* 0.4.9. All *p*-values < 0.05 were considered significant.

#### 4.6.2. Association of Congenital Malformations with Screening Variables

Association was tested using linear regression models, with the screening variables as dependent, congenital malformations as independent variable, and case-control status as covariate. Due to high skewness, screening variables were first log-transformed and then Z-standardized, while risk percentages were transformed into normally distributed data using inverse rank transformation. All *p*-values < 0.05 were considered significant.

#### 4.6.3. CNV Associations

Associations with CNVs were computed in *PLINK* [47]. We calculated burden tests, testing if the total number of detected CNVs differed between cases and controls or between carriers of congenital malformations and non-carriers. Due to the high number of tests, empirical *p*-values, which are then already corrected for multiple testing, based on 10,000 permutations, were computed. Empirical *p*-values < 0.05 were considered significant. Enrichment of CNV positions for GO terms and GWAS hits were tested using *FUMA* [48]. Positions of identified CNVs were not enriched for any GO terms or reported hits for neurodevelopmental disorders (at a false discovery rate (FDR) of 0.05).

#### 4.6.4. Differential Gene Expression

Analysis of differential gene expression was conducted in R [39]. Raw gene counts were *voom* transformed [49], and afterward, differential gene expression was calculated using the R-package *eBayes* function in the R-package *limma* [50]. Surrogate variable (SV) analysis was used to correct for possible batch effects as well as cell type heterogeneity [51]. For both CVS and term placental tissue, the first SV was detected as significant (according to the permutation procedure implemented in the package) and used as a covariate in the subsequent analyses. Furthermore, those variables that were significantly different between cases and controls, i.e., maternal age, maternal BMI in early pregnancy, as well as smoking, and parity (uniparous vs. multiparous), were used as covariates. We also included the child’s sex and gestational age (in weeks at sampling), as both have been associated with placental gene expression [52]. Differential gene expression was calculated for congenital malformations within the circulatory system and CNVs in both tissues. Within each analysis, multiple testing correction was applied based on the Benjamini–Hochberg [53] approach, and all *p*-values were considered significant at FDR of ≤0.1.

#### 4.6.5. *p*-Values

All reported *p*-values are two-sided. For the association of congenital malformations with screening variables, we hypothesized that individuals with congenital malformations were comparable to cases with regard to their outcome in screening variables. Hence, one-sided *p*-values are presented.

#### 4.6.6. Enrichment for Pre-Eclampsia Genes

Of the 15 genes that were differentially expressed in CVS tissue with congenital malformations within the circulatory, 11 genes had been associated with pre-eclampsia before. To assess if this overlap was nominal significantly higher than expected by chance, we assumed that 9% of all investigated genes are associated with pre-eclampsia, based on the results for Saei et al. [14]. Using a binomial distribution, the chance to get at least 11 hits out of 15 genes is 9.91 × 10^−11^.

## Figures and Tables

**Figure 1 ijms-23-11448-f001:**
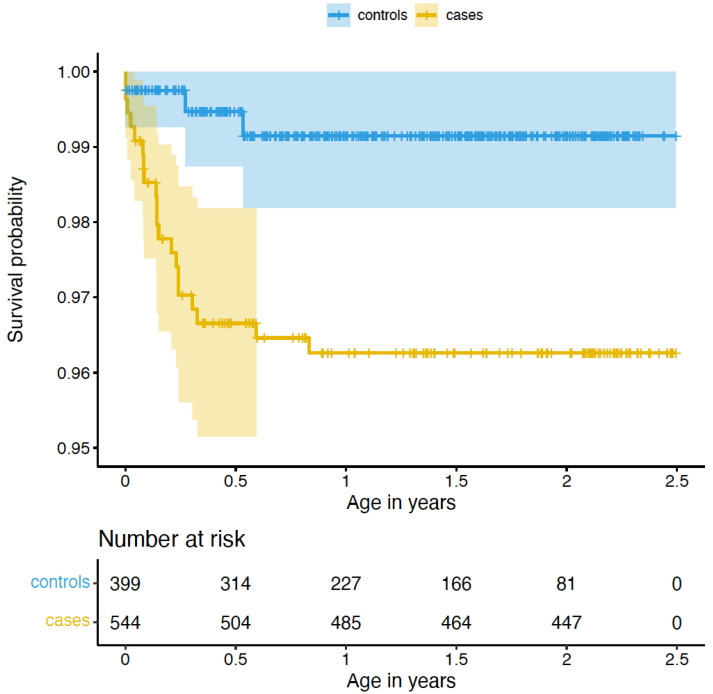
Survival curve for congenital malformations of the circulatory system compared between cases and controls. ‘Number at risk’ indicates the number of cases and controls available for analysis at 0–2.5 years of age. As cases presented with significantly longer follow-up time compared to controls, we limited the follow-up time of cases to the age of the eldest child in the control group (see also Methods).

**Figure 2 ijms-23-11448-f002:**
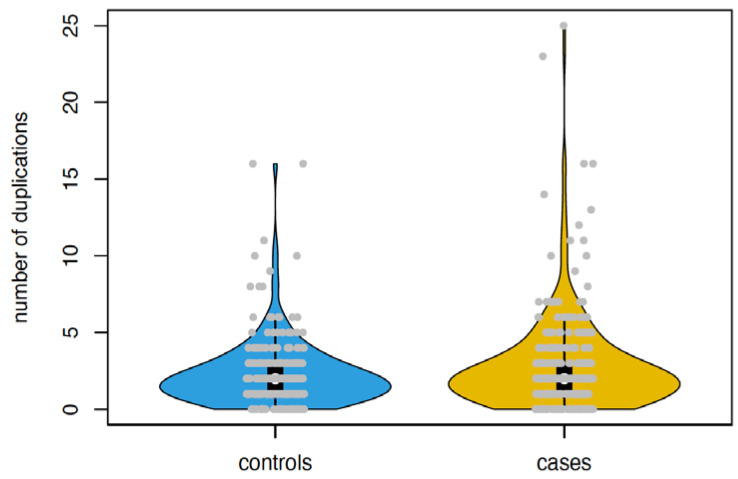
Differences in CNVs between cases and controls. Violin plot for numbers of called duplications between controls and cases. The difference remains significant after exclusion of two cases with more than 20 duplications (*p* = 0.03 based on 10,000 permutations).

**Figure 3 ijms-23-11448-f003:**
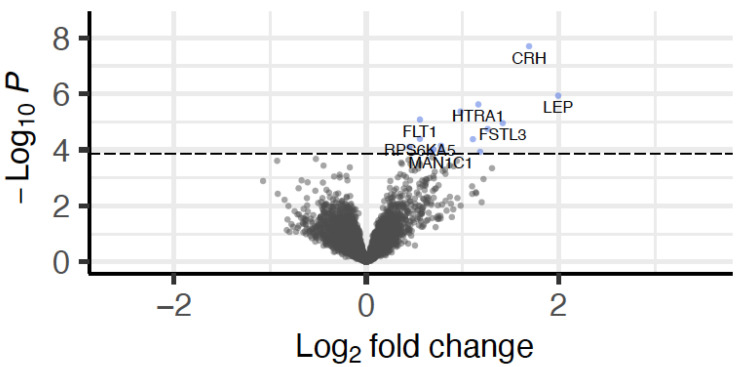
Volcano plot of differential gene expression in CVS with congenital malformations of the circulatory system. Genes differentially expressed at FDR of 0.10 are depicted in blue.

**Table 1 ijms-23-11448-t001:** Demographic information of the ITU cohort.

Maternal Characteristics	Cases (n = 544)	Controls (n = 399)	*p*-Value ^5^
Age at delivery, years, mean (SD), CI	35.54 (5.5) [35.52–35.56]	33.71 (4.2) [33.69–33.73]	**7.16 × 10^−9^**
Antenatal corticosteroid treatment, n (%)	23 (4.2%)	9 (2.3%)	1.41 × 10^−1^
Cesarean section, n (%)	120 (22.1%)	79 (19.8%)	4.48 × 10^−1^
Diabetes Disorders in pregnancy, n (%)	120 (22.06%)	77 (19.3%)	3.43 × 10^−1^
Early pregnancy BMI, median (IQR), CI	23.43 (5.23) [23.05–23.74]	22.65 (4.24) [22.27–23.03]	**1.42 × 10^−3^**
Hypertensive Disorders in pregnancy, n (%)	46 (8.5%)	20 (5.0%)	5.50 × 10^−2^
Primiparous, n (%)	221 (40.6%)	243 (60.9%)	**1.15 × 10^−9^**
Smoking during pregnancy ^1^, n (%)	36 (6.6%)	4 (1.0%)	**1.83 × 10^−5^**
Thyroid disorders ^2^, n (%)	9 (1.7%)	8 (2.0%)	8.79 × 10^−1^
**Child characteristics**			
Birth weight, g, median (IQR), CI	3533 (675) [3470–3585]	3562 (610) [3508–3620]	3.83 × 10^−1^
Gestational age at birth, weeks, median (IQR), CI	40.00 (2.0) [40.00–40.14]	40.14 (1.7) [40.00–40.29]	5.17 × 10^−1^
Preterm birth ^3^ (<37 weeks), n (%)	33 (6.1%)	13 (3.3%)	6.80 × 10^−2^
Sex, girl, (%)	256 (47.1%)	204 (51.1%)	2.42 × 10^−1^
**Screening variables**			
β-hCG level (MoM, ug/L), median (IQR), CI	1.31 (1.17) [1.30–1.73]	0.98 (0.74) [0.93–1.08]	**1.20 × 10^−11^**
NT, MoM mm, median (IQR), CI	1.80 (1.15) [1.58–1.88]	0.84 (0.32) [0.83–0.88]	**1.58 × 10^−36^**
PAPP-A level, MoM, mU/L, median (IQR), CI	0.75 (0.80) [0.75–1.08]	1.12 (0.73) [1.09–1.20]	**3.45 × 10^−18^**
Risk for Down syndrome ^4^, median (IQR), CI	0.65 (1.01) [0.63–0.89]	0.01 (0.2) [0.008–0.11]	**3.06 × 10^−101^**
Risk for trisomy 18 ^4^, median (IQR), CI	0.003 (0.01) [0.003–0.014]	0.001 (0.00) [0.001–0.001]	**1.15 × 10^−64^**

BMI: body mass index, calculated as weight in kg divided by height in meters squared. CI: confidence interval, CI for median was calculated as suggested in [9]. IQR: interquartile range. MoM: multiple of median. NT: nuchal translucency. SD: standard deviation. ^1^ Women who quit smoking in the first trimester were counted as non-smokers. ^2^ Thyroid disorders are based on ICD-10 codes E00-E07. ^3^ Preterm birth was defined as birth at gestational weeks < 37. ^4^ Risk for Down syndrome and for trisomy 18 were calculated as risk ratios based on PAPP-A and β-hCG levels, NT as well as maternal age, BMI and gestational weeks (see also Kvist et al. [8]). ^5^ Nominal *p*-value of testing phenotype in cases versus phenotype in controls, based on *t*-tests/Wilcoxon- tests for quantitative traits and on chi-square tests for categorical traits, nominally significant *p*-values < 0.05 are depicted in **bold**. Quantitative variables were checked for normality. If the Kolmogorov-Smirnow-test provided no indication for deviation from normality, mean and SD are reported and *p*-values are based on *t*-tests. If the variable was not normally distributed, median and IQR are reported and *p*-values are based on Wilcoxon tests.

**Table 2 ijms-23-11448-t002:** Self-reported maternal psychological health in cases and controls.

Phenotype	Cases (n = 260)	Controls (n = 353)	*p*-Value ^1^
CESD 19 gestational weeks, median (IQR), CI	9 (9) [9–14]	9 (8) [9–11]	0.32
CESD 26 gestational weeks, median (IQR), CI	8 (10) [9–12]	9 (8) [9–11]	0.35
CESD 38 gestational weeks, median (IQR), CI	8 (9) [8–11]	9 (9) [9–10]	0.43
Cohen’s perceived stress scale 19 gestational weeks, median (IQR), CI	5 (4.06) [6–8]	6 (4) [6–7]	0.48
Cohen’s 26 gestational weeks, median (IQR), CI	5 (5) [5–7]	5.5 (3) [6–6]	0.30
Cohen’s perceived stress scale 38 gestational weeks, median (IQR), CI	5 (4) [5–6]	5 (4) [4.5–6]	0.87
STAI 19 gestational weeks, median (IQR). CI	37 (11) [39–42]	38 (9) [38.95–41]	0.74
STAI 26 gestational weeks, median (IQR), CI	36 (9) [37–40]	37 (10) [37–39]	0.38
STAI 38 gestational weeks, median (IQR), CI	36 (10.42) [36–39]	37 (10) [36–38]	0.70

CESD: Center for Epidemiological Studies Depression. CI: confidence interval, CI for median were calculated as suggested in [9]. IQR: interquartile range. STAI: The State-Trait Anxiety Inventory. ^1^ *p*-value: Due to the non-normal distribution of the questionnaires, median and interquartile range (IQR) are presented, *p*-values are based on Wilcoxon tests.

**Table 3 ijms-23-11448-t003:** Differentially expressed genes in CVS with congenital malformations within the circulatory system.

Gene ^1^	Position (hg19) ^2^	Log2 (FC) ^3^	*p*-Value ^4^	Adjusted *p*-Value ^5^	Correlation CVS and Placenta ^6^
*LEP*	chr7: 127,881,331–127,897,682	1.99	1.16 × 10^−6^	5.28 × 10^−3^	r = 0.27; *p* = 0.01
*CRH*	chr8: 67,088,612–67,090,846	1.69	1.99 × 10^−8^	1.81 × 10^−4^	r = 0.14; *p* = 0.17
*FSTL3*	chr19: 676,389–683,392	1.42	1.11 × 10^−5^	1.68 × 10^−2^	r = 0.26; *p* = 0.01
*PAPPA2*	chr1: 176,432,307–176,811,970	1.26	1.79 × 10^−5^	2.32 × 10^−2^	r = 0.30; *p* < 0.01
*INHBA*	chr7: 41,733,514–41,818,976	1.18	1.18 × 10^−4^	7.36 × 10^−2^	r = 0.23; *p* = 0.03
*HTRA1*	chr10: 124,221,041–124,274,424	1.16	2.38 × 10^−6^	7.22 × 10^−3^	r = 0.30; *p* < 0.01
*DHRS2*	chr14: 24,105,573–24,114,848	1.11	4.16 × 10^−5^	4.20 × 10^−2^	r < 0.01; *p* = 0.97
*HS3ST3B1*	chr17: 14,204,506–14,249,492	0.98	4.27 × 10^−6^	9.71 × 10^−3^	r = 0.13; *p* = 0.23
*ANXA4*	chr2: 69,969,127–70,053,596	0.78	7.15 × 10^−5^	6.47 × 10^−2^	r < 0.01; *p* = 0.97
*MAN1C1*	chr1: 25,943,959–26,111,258	0.78	1.15 × 10^−4^	7.36 × 10^−2^	r = 0.05; *p* = 0.60
*MROH1*	chr8:145,202,919–145,316,843	0.70	9.64 × 10^−5^	7.30 × 10^−2^	r = −0.14; *p* = 0.17
*SEMA7A*	chr15: 74,701,630-74,726,299	0.68	1.21 × 10^−4^	7.36 × 10^−2^	r = 0.07; *p* = 0.53
*FLT1*	chr13: 28,874,483–29,069,265	0.56	8.27 × 10^−6^	1.50 × 10^−2^	r = 0.32; *p* < 0.01
*RPS6KA5*	chr14: 91,337,167–91,526,993	0.56	4.06 × 10^−5^	4.20 × 10^−2^	r = 0.30; *p* < 0.01
*HEXIM1*	chr17: 43,224,684–43,229,468	0.45	7.83 × 10^−5^	6.47 × 10^−2^	r < 0.01; *p* = 0.97

^1^ Gene: name of differentially expressed gene. ^2^ Position: gene position in hg19 coordinates. ^3^ Log2(FC): log2 (fold change) between cases and controls. ^4^ *p*-value: *p*-value for differential gene expression for congenital malformations within the circulatory system, adjusted for surrogate variable (SV), maternal age, maternal BMI in early pregnancy, smoking, parity, child’s sex, and gestational age at sampling. ^5^ Adjusted *p*-value: *p*-value after Benjamini–Hochberg correction for multiple testing (FDR). ^6^ Correlation: Pearson’s correlation coefficient and nominal *p*-value for correlation of gene expression between CVS and placental tissue (based on 93 individuals with gene expression available in both tissues and 7357 genes available in both tissues).

## Data Availability

Due to the sensitive nature of the patient data used in the current study and consent, the data sets are not and cannot be made publicly available. However, an interested researcher can obtain a de-identified data set after approval from ITU Study Board. Data requests may be subject to further review by the national register authority and by the ethical committees.

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
