# Peer review of "Genome-Wide Copy Number Variant and High-Throughput Transcriptomics Analyses of Placental Tissues Underscore Persisting Child Susceptibility in At-Risk Pregnancies Cleared in Standard Genetic Testing"

_ijms, 2022, doi:10.3390/ijms231911448_

Round 1
Reviewer 1 Report
Content suggestions:
The authors perfomed an interesting study, so I would like to kindly ask the authors only the following:
-
Do they have any data to provide the results of the impact of endocrine disorders of the mother in terms of thyreopathy and anatomical discrepancies of the uterus, tubae uterinae and ovaria on the pregnancy outcome ? This is relatively a frequently discussed question in this area of resarch.
From my perspective, the article can be published after incorporation of the answers to the questions and after such a minor revision.
Author Response
Response to Reviewer 1
The authors performed an interesting study, so I would like to kindly ask the authors only the following:
- Do they have any data to provide the results of the impact of endocrine disorders of the mother in terms of thyreopathy and anatomical discrepancies of the uterus, tubae uterinae and ovaria on the pregnancy outcome? This is relatively a frequently discussed question in this area of research.
From my perspective, the article can be published after incorporation of the answers to the questions and after such a minor revision.
Response: We want to thank the reviewer for the positive assessment of our study. Unfortunately, none of the pregnancy outcomes listed have been assessed specifically for this study in the mothers included. However, when accessing the Health register records for the participants, we were able to assess overall entries for thyroid disorders which are one of the most prevalent endocrine disorders in women (encoded with E00-E07 in ICD-10). Overall, 17 out of 943 women were diagnosed with thyroid disorders. As this proportion did not significantly differ between cases and controls, they did not change the outcome of our analyses. We now present this proportions also in Table 1.
Reviewer 2 Report
Manuscript: Genome-wide copy number variant and high-throughput transcriptomics analyses of placental tissues underscore persisting child susceptibility in at-risk pregnancies cleared in standard genetic testing
Objective:
The present study aimed to understand if children from pregnancies flagged for potential chromosomal aberrations but ultimately cleared for major aneuploidies following CVS or amniocentesis differed from children of inconspicuous pregnancies – and how the detection of prevailing pathologies could be improved. To explore this question, the authors used a thoroughly characterized perinatal cohort, the Finnish InTraUterine cohort (ITU), consisting of 943 mothers and babies followed throughout pregnancy and postnatally until 18 months. Also, the authors used genome-wide high-throughput analyses of cord blood's RNA and DNA methylation levels. In addition, the study demonstrated new findings being important in its area.
In this context, I had the following suggestions and corrections:
1. To include the email from all authors as recommended by the journal guidelines;
2. To exclude the topics from the abstract (background, methods, results, and conclusion;
3. Abstract:
To change “(n=544 cases) and 399 control pregnancies for “(n=544 cases, 57.7%) and control pregnancies (n=399 cases, 42.3%)”.
To exclude the term “(p=9.91x10-11)”. Only the citation of a significant association is essential in the abstract.
Edition: To exclude the “spaces” between the words. In some cases, there is more than one space between the words in the text.
Keywords: To cite the keywords using alphabetical order. Also, the authors can exclude the following keyword “copy-number variants” because the title had this word already.
4. Introduction
There are some paragraphs and sentences without references.
For example, in the following excerpt, “Although this screening test has proven to identify risk for the major chromosomal aneuploidies successfully, several studies have shown that fetuses with elevated risk at screening but cleared for the major aneuploidies in prenatal genetic testing remain at higher risk for negative outcomes diagnosed before or after birth.” The authors cited that several studies were previously published, and no citation was done about them.
5. Methods
In my opinion, the methods should be presented after the introduction. However, the journal guidelines recommended its presentation after the discussion. In the review, I will suggest the corrections in this session before the “Results” and “Discussion”
Table 1. Include the definition of BMI in the legend. The authors should consider the data distribution. The median and interquartile range is recommended for numerical data without normal distribution. In addition, I prefer to use mean (or median) with 95%CI to demonstrate the difference between the groups.
Statistical analysis
It is essential to inform the calculation for the different p-values cut-offs used in all analyses performed by the authors.
6. Results
In table 2 calls attention to the low adhesion of the participants from the case group. Is it able to compromise the findings (non-significative association)?
Figure 1. Excellent. The findings are clearly presented.
Figure 2. It is impressive the high number of CNVs in the control group. Please, it is essential to discuss these findings.
Table 3. It is important to describe the impact of an “r” varying from 0.23 to 0.30.
7. Discussion
To include the impact of the findings on a general population.
8. References
The authors used very old references. It is essential to cite new articles.
Minor comments
To revise the abbreviations in the study.
Author Response
Response to Reviewer 2
Comments:
- To include the email from all authors as recommended by the journal guidelines;
Response: We thank the reviewer for the positive assessment of our work. We have ensured that all email addresses are included.
- To exclude the topics from the abstract (background, methods, results, and conclusion;
Response: Thank you for noticing this – we have now removed the section headers.
- Abstract:
To change “(n=544 cases) and 399 control pregnancies for “(n=544 cases, 57.7%) and control pregnancies (n=399 cases, 42.3%)”.
To exclude the term “(p=9.91x10-11)”. Only the citation of a significant association is essential in the abstract.
Edition: To exclude the “spaces” between the words. In some cases, there is more than one space between the words in the text.
Keywords: To cite the keywords using alphabetical order. Also, the authors can exclude the following keyword “copy-number variants” because the title had this word already.
Response: We have made all of these edits.
- Introduction
There are some paragraphs and sentences without references.
For example, in the following excerpt, “Although this screening test has proven to identify risk for the major chromosomal aneuploidies successfully, several studies have shown that
fetuses with elevated risk at screening but cleared for the major aneuploidies in prenatal genetic testing remain at higher risk for negative outcomes diagnosed before or after birth.” The authors cited that several studies were previously published, and no citation was done about them.
Response: We have now included references for this sentence, as well as screening the rest of the manuscript for unreferenced sentences. We thank the reviewer for the detailed reading.
- Methods
In my opinion, the methods should be presented after the introduction. However, the journal guidelines recommended its presentation after the discussion. In the review, I will suggest the corrections in this session before the “Results” and “Discussion”
Response: Unfortunately, we cannot diverge from the journal format regarding the order of sections.
Table 1. Include the definition of BMI in the legend. The authors should consider the data distribution. The median and interquartile range is recommended for numerical data without normal distribution. In addition, I prefer to use mean (or median) with 95%CI to demonstrate the difference between the groups.
Response: We now present mean with 95% CI and t-tests for normally distributed variables and median with 95% CI and Wilcoxon-tests for non-normally distributed variables. We adapted Tables 1 and 2 accordingly.
Statistical analysis
It is essential to inform the calculation for the different p-values cut-offs used in all analyses performed by the authors.
Response: We apologize that this was unclear. We now give the different p-value thresholds in each sub-paragraph
- Results
In table 2 calls attention to the low adhesion of the participants from the case group. Is it able to compromise the findings (non-significative association)?
Response: It is true that the response rate is lower in case mothers, this could introduce a possible bias in the subgroup of women responding, who may be more or less affected than the whole group. Unfortunately we cannot test for such an ascertainment bias. We have now noted this limitation in the Discussion.
“However, it bares noting that there was a disproportionately lower adherence in the case mothers regarding the psychological self-assessment during pregnancy, which might influence the outcome of these analyses. Future studies should assess the impact of a similar exposure in additional cohorts.”
Figure 1. Excellent. The findings are clearly presented.
Response: Thank you for this positive assessment.
Figure 2. It is impressive the high number of CNVs in the control group. Please, it is essential to discuss these findings.
Response: While these numbers may seem high, our findings are in line with the literature. The average of 2.26 duplications per individual in controls as well as 2.8 per individual in cases is within the range of other studies. Across all CNVs (deletions and duplications), we identified an average of 6 CNVs per individual. This is also right on par with other studies: i.e. Redon et al (1) detected on average 5.4 CNVs per individual in n=270; Heo et al.(2) detected on average 5.16 total CNVs in n=300 individuals.
We have stated this in the manuscript as follows:
“In our cohort, cases presented with overall more duplications, but not more deletions, as compared to controls (burden analysis, based on all CNVs with at most 10% frequency, 823 called duplications in 297 cases versus 626 called duplications in 277 controls, p=0.01, based on 10,000 permutations, see Figure 2). These range of CNV counts is in line with previous studies using similar technology and sample sizes (1, 2).”
Table 3. It is important to describe the impact of an “r” varying from 0.23 to 0.30.
Response: We thank the reviewer for pointing this out. In fact, all differentially expressed genes which were positively correlated between CVS and placenta tissue, show only moderate to weak correlation. Hence, effects observed in CVS tissue cannot directly be translated to placenta.
We have stated this in the manuscript accordingly:
“Correlation of gene expression for these 15 genes between CVS and term placenta (based on 93 individuals who had both tissues available) revealed significant, albeit weak to moderate, positive correlations for 7 of these transcripts (r=0.23 to r=0.30, see Table 3).“
And also later in the discussion:
“15 genes showed differential expression association with circulatory system congenital malformations in first-trimester placenta obtained from CVS. Correlations with gene expression in placentas sampled at birth were weak to moderate, so we cannot directly translate our findings to late pregnancy.”
- Discussion
To include the impact of the findings on a general population.
Response: We have elaborated on this in the final Discussion paragraph as follows:
“The findings identified in this study join other reports suggesting that children with elevated risk but not meeting formal diagnostic thresholds during prenatal screening maintain a higher likelihood for negative outcomes. Increased awareness is key at the level of medical professionals with regards to potential early indicators of underlying disease susceptibility, and the prevention or intervention measures that can be taken to improve outcome and quality of life of affected individuals. In our cohort cases were significantly older as compared to control women which is in line what also Hayeems et al. reported(3). Women with higher age are more likely to undergo prenatal testing and higher maternal age has been associated with higher risk for fetal aneuploidies (4). Given that maternal age at first birth is shifting to older ages (5), this will likely be an issue of increasing importance. Our findings point to a need for re-defining the risk classifications in relation to prenatal screening, and more vigilance during prenatal and postnatal follow-up of at-risk children.”
- References
The authors used very old references. It is essential to cite new articles.
Response: We have adjusted some of the references to reflect newer perspectives. However, it is important to maintain some of the older references crediting the initial study that described for example the psychological questionnaires employed in the maternal stress evaluation (i.e. Radloff et al., Cohen et al., and Spielberger et al.) or statistical methods (i.e. ref Benjamini et al.)
Minor comments
To revise the abbreviations in the study.
Response: We thank the reviewer for spotting that we did not always follow the journal’s guidelines. We now adjusted abbreviations accordingly, i.e. defined them the first time they appear in each of three sections (abstract, main text, first figure/table).
1. Redon R, Ishikawa S, Fitch KR, Feuk L, Perry GH, Andrews TD, et al. Global variation in copy number in the human genome. Nature. 2006;444(7118):444-54.
2. Heo Y, Heo J, Han SS, Kim WJ, Cheong HS, Hong Y. Difference of copy number variation in blood of patients with lung cancer. Int J Biol Markers. 2021;36(1):3-9.
3. Hayeems RZ, Campitelli M, Ma X, Huang T, Walker M, Guttmann A. Rates of prenatal screening across health care regions in Ontario, Canada: a retrospective cohort study. CMAJ Open. 2015;3(2):E236-43.
4. Zhu H, Jin X, Xu Y, Zhang W, Liu X, Jin J, et al. Efficiency of non-invasive prenatal screening in pregnant women at advanced maternal age. BMC Pregnancy Childbirth. 2021;21(1):86.
5. Kim YN, Choi DW, Kim DS, Park EC, Kwon JY. Maternal age and risk of early neonatal mortality: a national cohort study. Sci Rep. 2021;11(1):814.
Reviewer 3 Report
Thank you to the authors for submitting this interesting manuscript. I have some minor queries:
1) Abstract - line 31 - please given additional detail about these "risk factors", even if just general
2) Abstract - line 43 - please give the estimation of association size between cases and copy number duplications compared to controls.
3) Line 93-96 - I recommend removing these sentences as they feel very much out of place given that the results follow immediately thereafter
4) Line 139 - what were the other types of malformations observed, even if not significantly different?
5) line 161-162 - please detail the pathogenic CNVs in a table, as well as the other CNVs
6) line 236-237 - I feel that this is **the** major take home from this study. I recommend highlighting/repeating this several times to heighten its recognition by the readership.
Author Response
Response to Reviewer 3
Thank you to the authors for submitting this interesting manuscript. I have some minor queries:
Response: We thank the reviewer for the positive assessment of our work.
1) Abstract - line 31 - please given additional detail about these "risk factors", even if just general
Response: We have edited the first Abstract sentence to read:
“Several studies have shown that children from pregnancies with estimated first trimester risk based on fetal nuchal translucency thickness and abnormal maternal serum pregnancy protein and hormone levels maintain a higher likelihood of adverse outcomes, even if initial testing for known genetic conditions is negative.”
2) Abstract - line 43 - please give the estimation of association size between cases and copy number duplications compared to controls.
Response: We adapted the abstract accordingly:
„Case children were significantly more often diagnosed with congenital malformations of the circulatory system (4.12 (95% CI [1.22-13.93]) higher hazard) and presented with significantly more copy number duplications as compared to controls (burden analysis, based on all copy number variants (CNVs) with at most 10% frequency, 823 called duplications in 297 cases versus 626 called duplications in 277 controls, p=0.01).“
3) Line 93-96 - I recommend removing these sentences as they feel very much out of place given that the results follow immediately thereafter
Response: We agree with the reviewer that the results follow immediately after, however, we also think that these lines highlight what the intention of this project was. Hence, we shortened the paragraph, indicating only the main message but without going into further details on the results in CVS tissue. These lines now read as follows:
“We used genome-wide high-throughput analyses at the level of RNA and DNA methylation in cord blood, term placentas, and, to the best of our knowledge, in the largest sample to date of CVS biopsies in the subset of pregnancies in which chromosomal testing was performed. Our findings point to an enduring potential for concern for these children, with increased risk for congenital malformations.”
4) Line 139 - what were the other types of malformations observed, even if not significantly different?
Response: Beyond malformations of the circulatory system, there were also children with malformations of the respiratory (n=2), musculoskeletal (n=20), genital (n=14), digestive (n=24), urinary (n=3) or nervous systems (n=1) as well as with facial malformations (n=11). These were either too infrequent (less than 1% of children affected) to permit statistical analysis or did not differ between cases and controls.
We have added the following sentence:
“Beyond malformations of the circulatory system, there were some isolated cases of malformations of the respiratory, musculoskeletal, genital, digestive, or nervous systems that were too infrequent to permit statistical analysis.”
5) line 161-162 - please detail the pathogenic CNVs in a table, as well as the other CNVs
Response: We thank the reviewer for this suggestion. We now present a detailed table of CNVs in the supplemental materials. The methods section now reads:
“This resulted in a final sample size of 574 IDs (277 controls and 297 cases) and 9,334 detected CNV calls including 3,445 CNVs (2,290 deletions and 1,155 duplications). We used ClassifyCNV (https://github.com/Genotek/ClassifyCNV) to classify detected CNVs according to guidelines(6) of the American College of Medical Genetics and Genomics into the categories: benign, likely benign, uncertain significance, likely pathogenic, or pathogenic. Of the called CNVs, 3,382 could be defined with ClassifyCNV, 239 CNVs were identified as likely pathogenic or pathogenic, 1,177 as benign and 1,966 with uncertain significance. A detailed list of classified CNVs is given in Suppl. Table 1.“
6) line 236-237 - I feel that this is **the** major take home from this study. I recommend highlighting/repeating this several times to heighten its recognition by the readership.
Response: We agree with the reviewer that one of the main messages of our manuscript is that common prenatal screening measurements can indicate the presence of additional risk for disease. While we already alluded to this in the abstract, we now also repeated this in the end of the discussion:
“The findings identified in this study join other reports suggesting that children with elevated risk but not meeting formal diagnostic thresholds during prenatal screening maintain a higher likelihood for negative outcomes. Increased awareness is key at the level of medical professionals as well as the general public in regards to potential early indicators of underlying disease susceptibility, and the prevention or intervention measures that can be taken to improve outcome and quality of life of affected individuals. In our cohort cases were significantly older as compared to control women which is in line what also Hayeems et al. reported(3). Women with higher age are more likely to undergo prenatal testing and higher maternal age has been associated with higher risk for fetal aneuploidies (4). Given that maternal age at first birth is shifting to older ages (5), this will likely be an issue of increasing importance. Our findings point to a need for re-defining the risk classifications in relation to prenatal screening, and more vigilance during prenatal and postnatal follow-up of at-risk children.”
- Redon R, Ishikawa S, Fitch KR, Feuk L, Perry GH, Andrews TD, et al. Global variation in copy number in the human genome. Nature. 2006;444(7118):444-54.
- Heo Y, Heo J, Han SS, Kim WJ, Cheong HS, Hong Y. Difference of copy number variation in blood of patients with lung cancer. Int J Biol Markers. 2021;36(1):3-9.
- Hayeems RZ, Campitelli M, Ma X, Huang T, Walker M, Guttmann A. Rates of prenatal screening across health care regions in Ontario, Canada: a retrospective cohort study. CMAJ Open. 2015;3(2):E236-43.
- Zhu H, Jin X, Xu Y, Zhang W, Liu X, Jin J, et al. Efficiency of non-invasive prenatal screening in pregnant women at advanced maternal age. BMC Pregnancy Childbirth. 2021;21(1):86.
- Kim YN, Choi DW, Kim DS, Park EC, Kwon JY. Maternal age and risk of early neonatal mortality: a national cohort study. Sci Rep. 2021;11(1):814.
- Riggs ER, Andersen EF, Cherry AM, Kantarci S, Kearney H, Patel A, et al. Technical standards for the interpretation and reporting of constitutional copy-number variants: a joint consensus recommendation of the American College of Medical Genetics and Genomics (ACMG) and the Clinical Genome Resource (ClinGen). Genet Med. 2020;22(2):245-57.